# miRNA Mediated Regulation and Interaction between Plants and Pathogens

**DOI:** 10.3390/ijms22062913

**Published:** 2021-03-13

**Authors:** Xiaoqian Yang, Lichun Zhang, Yuzhang Yang, Markus Schmid, Yanwei Wang

**Affiliations:** 1Beijing Advanced Innovation Center for Tree Breeding by Molecular Design, Beijing Forestry University, Beijing 100083, China; xiaoqianyang51@163.com (X.Y.); lczhang@bjfu.edu.cn (L.Z.); yuzhangyang0@163.com (Y.Y.); markus.schmid@umu.se (M.S.); 2National Engineering Laboratory for Tree Breeding, Beijing Forestry University, Beijing 100083, China; 3College of Biological Sciences and Biotechnology, Beijing Forestry University, Beijing 100083, China; 4Umeå Plant Science Centre, Department of Plant Physiology, Umeå University, SE-901 87 Umeå, Sweden

**Keywords:** miRNA, plant, pathogen, interaction, arms race

## Abstract

Plants have evolved diverse molecular mechanisms that enable them to respond to a wide range of pathogens. It has become clear that microRNAs, a class of short single-stranded RNA molecules that regulate gene expression at the transcriptional or post-translational level, play a crucial role in coordinating plant-pathogen interactions. Specifically, miRNAs have been shown to be involved in the regulation of phytohormone signals, reactive oxygen species, and *NBS-LRR* gene expression, thereby modulating the arms race between hosts and pathogens. Adding another level of complexity, it has recently been shown that specific lncRNAs (ceRNAs) can act as decoys that interact with and modulate the activity of miRNAs. Here we review recent findings regarding the roles of miRNA in plant defense, with a focus on the regulatory modes of miRNAs and their possible applications in breeding pathogen-resistance plants including crops and trees. Special emphasis is placed on discussing the role of miRNA in the arms race between hosts and pathogens, and the interaction between disease-related miRNAs and lncRNAs.

## 1. Introduction

MicroRNAs (miRNAs) are small non-coding RNA molecules of 20–24 nts in length which are encoded by *MIRNA* genes and can regulate complex biological processes in plants. *MIRNA* genes are usually transcribed by RNA polymerase II (pol II), and the initial product is a large primary miRNA (pri-miRNA) with a 5′-cap structure (7MGpppG) and a polyadenylated tail [1]. The pri-miRNA is processed to yield a precursor miRNA (pre-miRNA) with a self-complementary stem-loop. The pre-miRNA is diced to generate a miRNA/miRNA* duplex of approximately 22 nucleotides in length, which undergoes a methylation. The RNA duplex is then quickly exported into the cytoplasm where the mature miRNA interacts with Argonaute (AGO) protein to form the RNA-induced silencing complex (RISC) [2]. Alternatively, the mature miRNA might form a complex with AGO protein in the nucleus that is subsequently exported into the cytoplasm in a CRM1(EXPO1)/NES-dependent manner [3]. The N termini of all plant AGO1s contain a nuclear-localization (NLS) and nuclear-export signal (NES), which enables AGO1 nucleo-cytosolic shuttling [3]. Regardless of where in the cell the miRNA is loaded into the RISC, eventually, it binds by base pairing to a target RNA in cytoplasm [4,5,6].

The genetic regulation of miRNAs is one of the key mechanisms of plant response against biotic and abiotic stresses [7]. In general, most plant miRNAs guide AGO proteins to recognize their target RNAs by perfect or near-perfect sequence match and guide endonucleolytic target RNA cleavage, resulting in rapid degradation of target mRNA. miRNAs can also guide the cleavage of protein-coding and non-coding transcripts, such as those derived from TAS loci, to induce production of phased secondary small interfering RNAs (phasiRNAs), which in turn can lead to cleavage of complementary mRNAs [8,9,10]. miRNAs have also been shown to inhibit translation of their target mRNAs, thus limiting protein production [11,12,13]. Furthermore, miRNAs can regulate target gene expression by histone modification and DNA methylation [14,15]. For instance, 24-nt long miRNAs, called lmiRNAs, have been found to direct DNA methylation at their source and also function in trans on target genes to regulate gene expression in rice [16].

Plant diseases caused by various pathogens, such as bacteria, fungi, oomycetes, and viruses incur severe damage to forests and crop losses every year. Unraveling the interplay between miRNAs and their targets in disease resistance has been challenging. However, thanks to advances in high-throughput technologies, a growing number of miRNAs responding to biological stresses have been discovered and information regarding their expression levels, targets, and mode of action in the response to viruses, fungi, and bacteria in diverse plants is becoming clear. As a result, numerous regulatory circuits that place miRNAs in the process of plant defense against different pathogens have been identified in the past few years.

In this review, we provided an overview on current understanding concerning the role of functionally validated miRNAs in mediating the response to various pathogens across the different plant species at the molecular level. We summarized recent studies of the roles of miRNAs in plant defenses to elucidate the understanding of miRNA-pathogen interactions. The potential of miRNAs and their targets to enhance the resistance of crops and trees to pathogen through breeding or transgenic approaches are discussed.

## 2. Role of Plant miRNAs in Disease Resistance

Plants respond to pathogen invasion via two different types of immune responses, the pathogen-associated molecular pattern (PAMP)-triggered immunity (PTI) and effector-triggered immunity (ETI) [17,18]. Pathogens in turn can respond with a diverse array of virulence factors to suppress host defenses. Plant intracellular disease resistance (R) protein encoded by R genes can recognize these virulence effectors, often triggering a hypersensitive cell death response. Increasing evidence demonstrates that miRNAs are involved in PTI and ETI. In particular, miRNAs are involved in the regulation of a variety of defense signals and pathways, including *NBS-LRR* gene expression, hormone signals, reactive oxygen species (ROS) production, and cross-kingdom gene silencing, which contributes to the ongoing arms race between hosts and pathogens. Interestingly, miRNAs can also be decoyed by some specific ncRNAs (ceRNAs), thereby attenuating the repressive effect of miRNAs on their authentic targets. In the following, we will review the role of miRNAs in coordinating the responses of plants against pathogen attack and plant-pathogen interactions.

### 2.1. miRNA-Mediated Expression of Disease Resistance Genes

Plants have evolved diverse mechanisms to resist microbial pathogens, including regulation of defense genes by small RNAs. Interestingly, most of the resistance-related targets of miRNAs are *NBS-LRR* genes [19], which are characterized by a nucleotide-binding site (NBS) and a leucine-rich repeat (LRR). *NBS-LRR* genes constitute a central position in the innate defense system and are involved in ETI. miRNAs function as one of the master regulators of NBS-LRR defense gene family and target their conserved domains.

For example, miR482 has been shown to modulate potato resistance by suppressing *NBS-LRR* genes during *Verticillium dahlia* infection [20]. miR482 was also found to be involved in the pathogen response of cotton, where reduced expression of miR482 and increased the level of NBS-LRR transcripts were reported to confer resistance to *V. dahliae* [21]. Similar responses were also observed in tomato infected with *Fusarium oxysporum* [22]. In addition, poplar miR472a is involved in defense against *Colletotrichum gloeosporioides* and *Cytospora chrysosperma* by targeting NBS-LRR transcripts [23]. Considering the interaction between miRNAs and *NBS-LRRs*, a co-evolutionary model of plant *NBS-LRRs* and miRNAs has been proposed based on a large amount of miRNA data and *NBS-LRR* genes in 70 terrestrial plants, according to which the diversity of *NBS-LRR* genes can also promote the evolution of miRNAs, and more diverse miRNAs might be uncovered in future, further enriching the disease resistance regulatory network in plants [24].

One mode of action by which miRNAs can modulate *NBS-LRR* genes is through specific 22-nt miRNAs that trigger the biogenesis of phasiRNAs to amplify silencing efficiency on their targets [25,26] (Figure 1). For example, it has been reported that miR482 caused *NBS-LRR* mRNA decay and production of secondary siRNAs in diverse plants such as potato, tomato, and tobacco [5,20,27]. Similarly, 22-nt miR9863 was demonstrated to guide the cleavage of NLR-encoding gene *Mla* transcripts and trigger phasiRNAs during immune responses against the powdery mildew fungus in barley [28]. In legumes, miRNAs have also been shown to result in the production of phased trans-acting siRNAs [29]. Three 22-nt miRNA families (miR1507, miR2109, and miR2118) were identified as phasiRNAs generators and target conserved domains of *NBS-LRRs* [29]. Additionally, *Arabidopsis* miR472 and poplar miR472a could also trigger phasiRNAs to enhance *NBS-LRR* genes silencing [23,30].

### 2.2. miRNAs Involved in Phytohormone Signaling to Mediate Plant Immunity

Apart from modulating *NBS-LRR* function, miRNAs regulate the response of plants against pathogens also by more indirect routes. Several plant hormones including auxin, salicylic acid (SA), jasmonic acid (JA), and ethylene have been shown to modulate inducible defense reactions in response to pathogen attack. Interestingly, miRNAs seem to play a key role in mediating these responses. Evidence for a role of miRNAs in attenuating plant hormone signaling pathways to modulate defense reactions was first reported in *Arabidopsis* [31]. Perception of the bacterial flagellin flg22 causes an increase in miR393 levels, which negatively regulates transcripts encoding F-box auxin receptors TIR1, AFB2, and AFB3. The resulting repression of auxin signaling results in the resistance of *Arabidopsis* to *P. syringae* pv *tomato* DC3000 [31]. miRNA-mediated auxin regulation also plays an important role in virus aphid transmission (Vat)-mediated resistance against *Aphis gossypii* in melon. More specifically, target genes of miR393 encoding auxin receptors are suppressed in response to aphid infestation in *Vat^+^* resistant plants, corresponding to the induction of miR393 [32]. Furthermore, miR160 has been reported to regulate the immunity of potatoes against *Phytophthora infestans* by targeting genes encoding AUXIN RESPONSE FACTORS 10 (StARF10) and StARF16 [33]. Interestingly, both miR160 overexpression and knockdown lines in potato exhibited increased susceptibility to *P. infestans* compared with wild type, suggesting that overexpression or knockdown of miR160 may break the balance between antagonistic auxin and SA signaling pathways and that maintaining moderated miR160 levels might be crucial for defense responses in wild-type plants [33]. Of course, auxin signaling is not the only place where miRNAs have been implicated in pathogen responses. For example, Can-miRn371 mediates suppression of genes encoding ETHYLENE RESPONSE FACTORs (ERFs) and contributed to the resistance of chili against anthracnose pathogen *Colletotrichum truncatum*. Supporting this notion is the finding that the expression of three ERFs is down-regulated in resistant chili cultivars as well as in transgenic lines overexpressing Can-miRn37a in otherwise susceptible cultivars [34].

### 2.3. miRNAs Involved in ROS to Mediate Plant Immunity

In plants, the production of reactive oxygen species (ROS) such as superoxide radicals (O_2_^−^), hydroxyl radicals(^−^OH), and hydrogen peroxide (H_2_O_2_) is regarded as an important defense reaction for combating biotic and abiotic stresses [35]. miRNAs have been implicated in regulating ROS in response to various pathogens. In rice, miR398b has been reported to promote superoxide dismutase (SOD) activity in response to fungus *Magnaporthe oryzae* invasion, thereby increasing H_2_O_2_ concentration and plant resistance [36]. Accumulation of H_2_O_2_ was also detected in leaves of poplar transgenic lines overexpressing miR472a and STTM472a, and non-trangenic plants treated with a pathogen. However, reduced ROS accumulation was observed in transgenic poplar lines overexpressing miR472a compared with that in the non-transgenic poplar and lines expressing STTM472a [23].

### 2.4. miRNAs* Involved in Exocytosis and Other Pathways to Mediate Plant Immunity

Another level at which small RNAs have been shown to modulate immune responses is through the regulation of secretory pathways. In particular, it has shown that small RNAs can regulate plant immunity by regulating exocytosis [37]. As mentioned above, mature miRNAs are derived from a miRNA/miRNA* duplex. Originally, it was believed that only the miRNA could regulate downstream targets whereas the miRNA* was considered as a by-product that was rapidly degraded during the process of miRNA biogenesis [38]. Interestingly, in many of the cases in which small RNAs regulate secretory pathways, it seems to be the miRNA* rather than the miRNA itself that mediates these effects. For example, *Arabidopsis* miR393b* is loaded into AGO2 to modulate exocytosis of antimicrobial pathogenesis-related (PR) proteins in innate immunity via miRNA393*-mediated silencing of the Golgi-localized SNARE gene *MEMB12* [37]. This finding provides an example of a functional miRNA*, miR393b*, which promotes secretion of PR proteins and regulates ETI. Interestingly, the cognate miRNA miR393 has been shown to target transcripts encoding auxin receptors (see above), thereby contributing to PTI. Additional examples that implicate miRNA*s in biotic interactions include miR825*, which targets *TIR-NBS-LRR* genes to negatively regulate plant innate immunity [39], and some functional miRNA*s, which were reported to participate in the interaction between plants and arbuscular mycorrhizal (AM) fungi [40]. Based on these findings it would appear as if miRNA*s may play a particularly important role in plants. It would be interesting and meaningful to investigate the biological function of miRNA* in plants in more detail.

## 3. Role of Small RNAs including miRNAs in the Arms Race between Host and Pathogen

Plants will inevitably encounter a variety of biotic stresses during growth. A consequence of the constant exposure is an ongoing arms race in which plants constantly try to evolve mechanisms to counter potentially harmful microorganisms while the latter tries to overcome these protective measures. At the core of this ongoing battle are so-called effector proteins that are produced by pathogens to subvert host immune responses and facilitate disease development [41]. Some pathogen effectors act by inhibiting the biogenesis of small RNAs, thereby suppressing RNA silencing in plants [42]. Phytophthora Suppressor of RNA Silencing 1 (PSR1) can bind to the nuclear protein PSR1-Interacting Protein 1 (PINP1), which affects the localization of the Dicer-like 1 protein complex, leading to impaired miRNA-processing [41]. Recent studies have found that some pathogens can deploy cross-kingdom small-RNA effectors that attenuate host immunity and facilitate infection. Interestingly, host plants sometimes respond to attack by exporting specific sRNAs including miRNAs to induce cross-kingdom gene silencing in pathogenic fungi, thereby conferring disease resistance (Figure 2) [43,44]. The disease-related cross-kingdom sRNAs known so far acting in different plants and pathogens are summarized in Table 1.

For example, infection with *Phytophthora* induces the production of a large number of secondary siRNAs from specific transcripts in *Arabidopsis*. These siRNAs enter the *Phytophthora* via extracellular vesicles and silence specific *Phytophthora* target genes to confer resistance [45]. Similarly, in response to infection with *V. dahliae*, *Arabidopsis* miR166 was exported into the fungal hyphae to suppress pathogenicity [46]. miR166 and miR159 were also induced in cotton and exported to the pathogenic hyphae to inhibit virulence gene expression in response to *V. dahliae* infection [43]. Two *V. dahliae* genes *Clp-1* and *HiC-15*, which are essential for fungal virulence, are targeted by miR166 and miR159, respectively. This cross-kingdom inhibition of virulence gene in pathogenic fungi confers cotton disease resistance [43]. Potentially even more interesting is the interaction between *B. cinerea* and *Arabidopsis*. In this case, *B. cinerea* was reported to transfer some small RNAs (Bc-sRNAs) into host plant cells and hijack the host RNA interference (RNAi) machinery by binding to *Arabidopsis* ARGONAUTE 1 (AGO1) and selectively silencing host disease resistance genes to suppress host immunity [47]. Conversely, *Arabidopsis* secreted exosome-like extracellular vesicles to deliver sRNAs including miR166 into *B. cinerea* [46], making this a prime example of an arms race in which both sides, plants and pathogens, employ small RNAs.

In plants, an active immune response usually has deleterious effects on plant growth, hence maintaining the balance between yield and disease resistance has become a major challenge in plant breeding [48,49]. miRNA-mediated R gene turnover has been proven to be a protective mechanism for plants to prevent autoimmunity in the absence of pathogens. It has recently been demonstrated that miR1885 is involved in balancing the tradeoff between growth and defense in *Brassica* through distinct modes of action. miR1885-dependent silencing of the photosynthesis-related gene *BraCP24* was induced upon *Turnip mosaic virus* (TuMV) infection, speeding up floral transition, whereas miR1885-mediated turnover of the R gene *BraTNLI* was overcome by TuMV-induced BraTNLI expression [48]. The precise and dynamic modulation of the interplay between growth, immunity, and pathogen infection reflects the sophisticated arms race between plants and pathogens.

## 4. Interaction between Disease Related miRNA and lncRNA

An interesting feature of miRNAs is that their abundance is not only regulated at the level of transcription and processing of the miRNA precursor but though other RNA molecules that directly interact with and inhibit miRNAs though (partial) sequence complementarity. Interestingly, it has recently been shown that some miRNAs could potentially be decoyed by some specific endogenous long non-coding RNAs (lncRNAs) (Figure 2). LncRNAs are a class of RNA transcripts (>200 nt) lacking protein-coding potential and have lower sequence conservation compared with miRNAs. Such lncRNAs with miRNA complementarity could act as endogenous target mimics (eTMs), also referred to as ceRNAs (competing endogenous RNAs), to decoy miRNAs by competing for their targets, thereby attenuating the repressive effect of miRNAs on their targets [53,54,55,56,57,58]. Numerous lncRNAs have been predicted as miRNA decoys to regulate plant immunity through transcriptome sequencing and co-expression analysis in maize, wheat, melon, and tomato [59,60,61,62]. While only a few lncRNAs-miRNAs interactions were verified by transgenic methods in tomato. lncRNA23468 functions as a ceRNA that modulates *NBS-LRR* genes by decoying miR482b during *Phytophthora infestans* infection in tomato, thus establishing a lncRNA23468-miR482b-*NBS-LRR* network of tomato resistance to *P. infestans* [63]. Similarly, miR159 was decoyed by lncRNA42705 and lncRNA08711 resulting in increasing expression level of its target MYBs [61], and lncRNA39026 could function as ceRNA of miR168a to modulate *PR* genes in tomato for enhanced resistance to *P. infestans* [64]. LncRNAs could act as miRNA decoys in response to pathogen infection, which adds complexity at the level of RNA interaction regulation in the plant. Taken together, while the study of lncRNAs in modulating plant-pathogen interactions is still at its beginning, this particular line of research holds a lot of promise for the future.

## 5. Regulatory Modes of Disease-Related miRNAs in Plants

Plants have evolved diverse molecular mechanisms that enable them to respond to a wide range of pathogens. Disease-related miRNAs that are involved in a variety of defense signals and pathways and that can affect plant immunity either positively or negatively, are summarized in Appendix A [5,20,21,22,23,27,31,34,35,36,37,38,39,43,45,65,66,67,68,69,70,71,72,73,74,75,76,77,78,79,80,81,82,83,84,85,86,87]. The main targets of these miRNAs are *NBS-LRR* genes, hormone receptors, transcription factors, and superoxide dismutase (SOD) family genes. Of these, miRNAs targeting hormone receptors regulate defense response mainly in a positive manner, whereas miRNAs targeting *NBS-LRR* transcripts principally play a negative role in plant immunity (Appendix A).

Although many miRNAs are conserved in plants, their expression levels, targets, and regulation patterns change to varying degrees in the defense process of plants to viruses, fungi, and bacteria among different plant species. For example, miR398b was reported to positively regulate the immunity response of rice against *M. oryzae* [36], whereas miR398b plays a negative role in the defense response of *Arabidopsis* against DC3000 bacteria [67,78], indicating that miR398b may be involved in immunity responses with an inverse regulatory mode of action for bacterial and fungal pathogen species. This may be related to an integrative regulatory module mediated by miR398b, which targets genes encoding superoxide dismutase (SOD) family members CCSD, CSD, and SODX [36,78]. Antagonistic regulation of one miRNA was also observed in poplar where miR472a overexpression lines showed higher susceptibility to flg22 and hemibiotroph *Colletotrichum gloeosporioides* but exhibited enhanced resistance to the necrotrophic fungus *Cytospora chrysosperma* [23]. Similarly, tomato miR482, whose targets certain *NBS-LRR* genes was repressed in the defense reaction to bacteria and viruses [5], whereas poplar miR164 and miR1448, which also target *NBS-LRR* genes, increased during canker pathogen infection [72]. Thus, different pathogen species may affect miRNA regulation differently. It may be related to the effector types secreted by different pathogens, the specific recognition of different pathogen effectors by plant R genes which can be regulated by diverse miRNAs plays a key role in the immune response in plants. Apart from that, this difference may be caused by the stress strength of the pathogen infection, tomato was sampled at 4 h post-inoculation whereas poplar was sampled at more than 72 h (3, 5, and 7 days) post-inoculation in above research [5,72], indicating miRNAs expression might display at the dynamic expression changes of immunity response. Pathological development is a dynamic process. Therefore, research conclusions are sometimes subject to different sampling experiment designs for the inoculation stages.

Plants developed a suite of defensive mechanisms to cope with environmental stresses. Accordingly, miRNAs grouped in the same family might have evolved more members in different species or varieties and some plants have also evolved some species-specific miRNAs coordinating the complex regulatory mechanisms in plants [88]. miR482/2118 superfamily is one of the conserved disease-related miRNA families in cotton, *Populus,* and *Solanaceae*, but this family does not exist in *Arabidopsis* [89,90,91]. Several newly expanded MIR482/2118d loci have mutated to produce different miR482/2118 variants with altered target-gene specificity in tetraploid cotton compared with their extant diploid progenitors [91]. Previous research showed that the resistance gene *R3a* in potato was cleaved by miR482 family and produced phasiRNA [86]. In contrast, in tomato, the *R3a* homolog *I2* was targeted by miR6024 even though miR482 also exists [86]. Evolutionary analysis of *I2* homologs revealed a considerable divergence between potato and tomato *I2* locus, which may account for the regulation of *I2* homologs by two miRNAs [86]. Hence, evolution of resistance gene families may attribute to the regulation mediated by different miRNAs.

## 6. The Applications of miRNAs in Molecular Breeding for Disease Resistance

The elucidation of miRNA function in the regulation of target genes has led to the development and application of several miRNA-based approaches for plant breeding. This is also the case in plant disease-resistant breeding, where understanding the role of specific miRNAs in plant-pathogen interaction systems has already led to several successful applications. One straightforward method is to modulate agronomic traits by constitutively overexpressing a specific miRNA. For example, overexpression of rice-specific miRNA osa-miR7695 increases resistance to infection by the fungal pathogen *M. oryzae* [68]. On the other hand, overexpressing a miRNA-resistant form of the target that escapes the cleavage by its miRNA or using an artificial miRNA–target mimic that inhibits the activity of a given miRNA might also be effective strategies to attenuate the effect of miRNAs that function as negative regulators of the traits of interest [53]. Recently, artificial miRNAs (amiRNAs) have become increasingly popular in plant disease resistance breeding. In particular when it comes to battling the effects of the virus, which are mostly mixed infections and mutate rapidly, on the productivity of crops and trees. Intriguingly, amiRNAs to a certain extent allow mismatches in their target region. Artificial miRNAs designed to target relatively conserved regions of a virus can effectively inhibit most virus strains and confer resistance even after the virus has started to accumulate mutations [92,93], which is of great significance for the cultivation of durable and broad-spectrum antiviral varieties. For instance, transgenic tomato plants expressing an artificial miRNA targeting the ATP/GTP binding domain of *AC1* gene of tomato leaf curl New Delhi virus (ToLCNDV) could effectively resist ToLCNDV infection [94]. Artificial miRNAs were also successfully used to effectively inhibit the infection of tobacco plants by *Potato virus Y* (PVY^N^) and *Tabacco etch* virus (TEV) [92].

While the transgenic approaches obviously work, such plants might be difficult to market. Hence other technologies are needed. The latest developments in genome editing tools have paved the way for targeted mutagenesis, opening new horizons for precise genome engineering. For example, CRISPR/Cas9 is an entrancing and versatile tool for plant genome editing [95], as it enables genetic improvement by editing endogenous genes. In the best scenario, such improvements are made using DNA-free delivery of CRISPR/Cas9, thus completely avoiding the formation of transgenic plants [95,96]. But even if transgenes are introduced in the plant genome to facilitate efficient mutagenizes, these transgenes are usually in trans to the targeted locus and can be removed by genetic crossings, thus reducing safety issues that are a major public concern when it comes to traditional transgenic methods. One way by which CRISPR/Cas9 can be employed is to directly disrupt disease-causing genes and develop disease-resistant crops. For example, targeted knockout of the ethylene-responsive gene *OsERF922* using CRISPR/Cas9 resulted in increased resistance against *Magnaporthe oryzae* in rice [97].

Likewise, CRISPR/Cas9 could be employed to mutate *MIRNA* genes to enable the plant to generate novel miRNAs that, for example, target pathogen effectors normally not recognized by the plant. Alternatively, a strategy could also be employed to introduce silent mutations in transcripts that are targeted by small RNAs originating from pathogens (see above). However, while such an approach seems feasible in principle, it has not been reported in plants so far. In contrast, several cases of successful CRISPR/Cas-mediated engineering of *cis*-regulatory elements via genome editing in plants have been reported. For example, the deletion of a regulatory fragment containing a transcription-activator-like effector (TALe)-binding element (EBE) through CRISPR/Cas9 in the promoter of *SWEET11* improved rice disease resistance [98,99]. Furthermore, editing of effector-binding elements (EBEs) in the promoters of *SWEET* genes resulted in rice lines with bacterial blight resistance [98,99]. In principle, mutations might also be induced in *cis*-regulatory regions of disease-related *MIRNAs* to regulate their expression. However, to the best of our knowledge, such experiments have not yet been reported. These examples clearly demonstrate the potential of CRISPR/Cas9 for engineering resistance traits in crops. The challenges that remain are mainly to improve the efficiency and fidelity of CRISPR/Cas9 mutagenesis in a wide range of plant materials.

## 7. Perspectives

miRNAs have been shown to modulate plant immune responses at various levels as regulation of gene expression by miRNAs is a key mechanism that facilitates the response of plants to biotic stresses. However, the situation is more complex than simple one-to-one relations. miRNAs are central components of complex regulatory networks in which an individual miRNA may target more than one transcript and vice versa to modulate and fine-tune expression of the genome. Therefore, studying miRNA-target interactions provides valuable insights into mechanisms of transcriptional or post-transcriptional gene regulation in general and the multiple molecular pathways that control plant stress responses in particular [100]. The main targets of the miRNAs when it comes to the regulation of plant immune responses are *NBS-LRR* genes, TFs, and hormone receptors, among others. A theme that is emerging from these studies is that the regulation of disease-related miRNAs and their effect on the progression of an infection is very much dependent on the plant and pathogen species. Some miRNAs were regulated in an antagonistic manner between different plants, or in the same plant in response to different pathogens. A take-home message from these studies is that even for evolutionarily conserved miRNAs and their targets, one should not assume that the regulatory mechanisms and their role in plant defense against pathogens are equally conserved.

An area of research that will need to be addressed in more detail in the future concerns the role of miRNAs in the arms race between hosts and pathogens. For example, the mechanisms by which fungal pathogens secrete and deliver small RNA effectors to the plant host as well as the host exports sRNAs to induce cross-kingdom gene silencing still remain poorly understood. Furthermore, future research into the biological function of miRNA*s, which were originally considered a byproduct of miRNA biogenesis, will enhance our understanding of the regulatory function of miRNA networks in plant innate immunity. miRNAs and their targets are also offering opportunities for developing novel strategies and technologies to improve pathogen resistance in crops. For example, amiRNAs technology has been gradually adopted increasingly in land plant disease studies. Recent developments in genome editing are also started to be used in engineering miRNAs and their targets to breed pathogen-resistant crops and we can expect to see more of this in the future. In particular, CRISPR/Cas9 could be used to mutate miRNAs that negatively regulate plant resistance. Alternatively, mutations could also be induced in *cis*-regulatory regions of disease-related MIRNA genes to regulate their expression. Undoubtedly, the identification of additional miRNA-target modules, as well as the application of novel genome editing tools, will make miRNAs a focus of resistance breeding in crops and trees in the future [101].

## Figures and Tables

**Figure 1 ijms-22-02913-f001:**
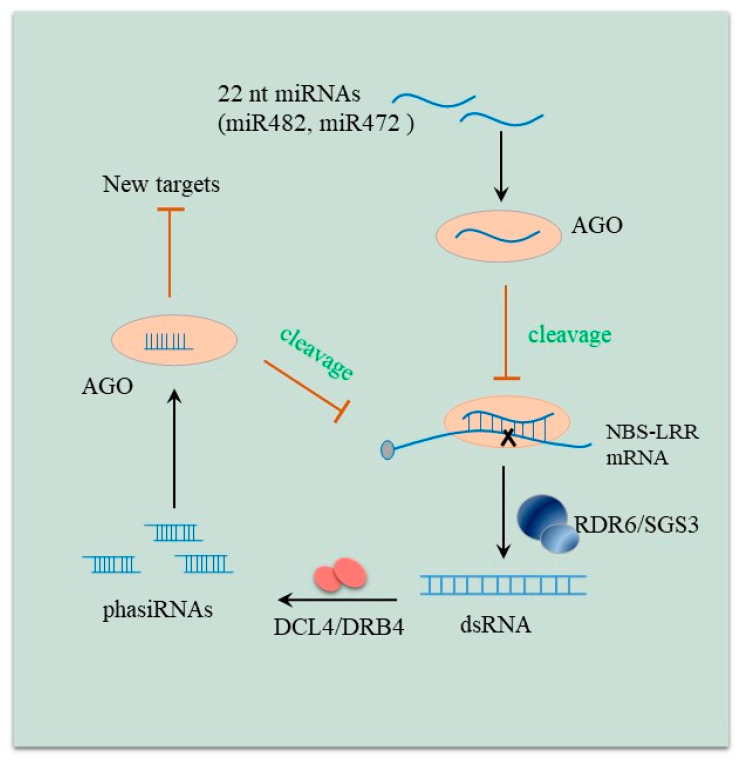
The regulatory network of 22 nucleotides (nt)-long miRNAs and *NBS-LRR* mRNAs involved in production of phasiRNAs. **The** 22-nt miRNA guides AGO protein to cleave the target site on the *NBS-LRR* transcript, triggering dsRNA synthesis mediated by RDR6 (RNA-DEPENDENT RNA POLYMERASES 6) and SGS3 (SUPPRESSOR OF GENE SILENCING 3). dsRNA is subsequently processed by DCL4 (DICER-LIKE 4) and DRB4 (DOUBLE-STRANDED-RNA-BINDING PROTEIN 4) to generate a cluster of 21-nt phased siRNAs (phasiRNAs). These 21-nt phasiRNAs are loaded into AGO proteins, which in turn can lead to *NBS-LRR* mRNAs cleavage. On the other hand, these siRNAs will depress more new targets.

**Figure 2 ijms-22-02913-f002:**
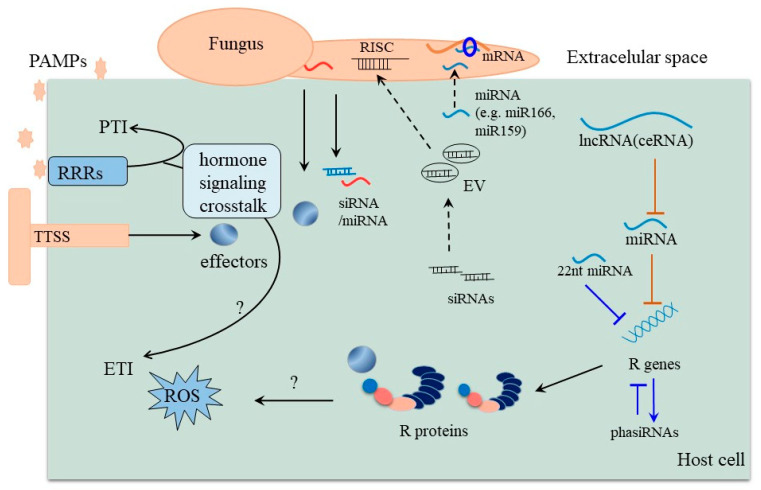
Hosts and pathogens are engaged in an arms race mediated by sRNAs. In plant–pathogen interactions, plants recognize PAMPs (pathogen-associated molecular pattern) with PRRs (pattern recognition receptor) to trigger innate immunity against most pathogen infections. Pathogens, in turn, secrete effectors including effector proteins and some sRNAs to facilitate infection. Some plant NB-LRR proteins are activated by effector proteins, triggering the second layer of immunity response, the so-called ETI. Additionally, hosts simultaneously export siRNAs and miRNAs to induce cross-kingdom gene silencing in the pathogen. Some plant miRNAs targeting disease resistance genes can be decoyed by certain specific lncRNAs, which attenuates the repression of miRNAs on their targets. EV: extracellular vesicles; TTSS: type Ⅲ secretion system. (Question symbols indicate predicted characteristics that need further mining).

**Table 1 ijms-22-02913-t001:** Summary of disease-related cross-kingdom sRNAs acting in different plants and pathogens.

Plant sRNA (Pathogen sRNA)	Targets of Plant sRNA in Pathogen	Plant	Pathogen	Referencs
miR166		*Arabidopsis thaliana*	*Botrytis cinerea*	[46]
miR1023	*FGSG_03101*	Wheat (*Triticum aestivum*)	*Fusarium graminearum*	[50]
miR166	*Clp-1*	Cotton (*Gossypium hirsutum*)	*Verticillium dahliae*	[43]
miR159	*HiC-15*	Cotton (*Gossypium hirsutum*)	*Verticillium dahliae*	[43]
(*Pst*-milR1)		Wheat (*Triticum aestivum*)	*Puccinia striiformis* f. sp.*tritici* (*Pst*)	[51]
TAS1c-siR483	*BC1G_10728; BC1G_10508*	*Arabidopsis thaliana*	*Botrytis cinerea*	[46]
TAS2-siR453	*BC1G_08464*	*Arabidopsis thaliana*	*Botrytis cinerea*	[46]
IGN-siR1		*Arabidopsis thaliana*	*Botrytis cinerea*	[46]
*Bc*-*DCL*-targeting sRNAs	*Bc*-*DCL*	*Arabidopsis thaliana*	*Botrytis cinerea*	[52]
siRNA-1310	*Phyca_554980*	*Arabidopsis thaliana*	*Phytophthora*	[45]
(Bc-siR3.1)		*Arabidopsis thaliana;* Tomato (*Solanum lycopersicum*)	*Botrytis cinerea*	[47]
(Bc-siR3.2)		*Arabidopsis thaliana;* Tomato (*Solanum lycopersicum*)	*Botrytis cinerea*	[47]
(Bc-siR5)		*Arabidopsis thaliana;* Tomato (*Solanum lycopersicum*)	*Botrytis cinerea*	[47]

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
