# Peer review of "miRNA Mediated Regulation and Interaction between Plants and Pathogens"

_ijms, 2021, doi:10.3390/ijms22062913_

Round 1

Reviewer 1 Report

Yang and coauthors present a review about a subject that needs to be often updated, since new findings and increased knowledge emerge continuosly. miRNAs are well known regulators of plant-pathogens interactions. New NGS studies and new breeding technologies (e.g. genome editing), however, are providing a wealth of new data that still need to be fully understood. Many new mechanisms are being unveiled, and the present manuscript helpfully revisit and summarize the current knowledge on how those mechanisms interfere with plants susceptibility/resistance responses to their pathogens. The deriving review is comprehensive on the topic and will surely be of interest for the readers of this journal.

Please find some minor revisions throughout the text:

Line 33: Polyadenylated tail

Line 86: ... in coordinating the responses (without THE)

Line 103: Sentence interrupted before the full stop. Please rewrite

Line 112: In reference [28], barley miR9863 is shown to trigger phasiRNAs and regulate Mla genes transcript cleavage and degradation. Therefore the sentence "miR9863 attenuated phasiRNAs mediating NBS-LRRs cleavage" is misleading and should be reworded.

Line 282: "LncRNAs could (...) as miRNA decoys in response to pathogen infection": in this sentence a verb is missing.

Lines 379-80: "such improvements are made using DNA-free delivery of CRISPR/Cas9...": please add a reference.

Author Response

Response to Reviewer 1 Comments

     Yang and coauthors present a review about a subject that needs to be often updated, since new findings and increased knowledge emerge continuosly. miRNAs are well known regulators of plant-pathogens interactions. New NGS studies and new breeding technologies (e.g. genome editing), however, are providing a wealth of new data that still need to be fully understood. Many new mechanisms are being unveiled, and the present manuscript helpfully revisit and summarize the current knowledge on how those mechanisms interfere with plants susceptibility/resistance responses to their pathogens. The deriving review is comprehensive on the topic and will surely be of interest for the readers of this journal. Please find some minor revisions throughout the text:

    Response: We are very grateful to your comments for our manuscript. According to your advice, we have modified the relevant parts in the revised version. We have uploaded the revised manuscript using the "Track Changes" in the formats of MS word for your approval.

Point 1: Line 33: Polyadenylated tail

Response 1: We have modified “Polyadenylation tail” to “Polyadenylated tail” as suggested in the revised manuscript.

Point 2: Line 86: ... in coordinating the responses (without THE)

Response 2: We have deleted “the” as suggested in the revised manuscript.

Point 3: Line 103: Sentence interrupted before the full stop. Please rewrite

Response 3: The sentence has been rewritten as suggested in the revised manuscript:

Considering the interaction between miRNAs and NBS-LRRs, a co-evolutionary model of plant NBS-LRRs and miRNAs has bene proposed based on a large amount of miRNA data and NBS-LRR genes in 70 terrestrial plants, according to which the diversity of NBS-LRR genes can also promote the evolution of miRNAs, and more diverse miRNAs might be uncovered in future, further enriching the disease resistance regulatory network in plants.

Point 4: Line 112: In reference [28], barley miR9863 is shown to trigger phasiRNAs and regulate Mla genes transcript cleavage and degradation. Therefore, the sentence "miR9863 attenuated phasiRNAs mediating NBS-LRRs cleavage" is misleading and should be reworded.

Response 4: The sentence has been rewritten  as suggested in the revised manuscript:

Similarly, 22-nt miR9863 was demonstrated to guide the cleavage of NLR-encoding gene Mla transcripts and trigger phasiRNAs during immune responses against the powdery mildew fungus in barley.

Point 5: Line 282: "LncRNAs could (...) as miRNA decoys in response to pathogen infection": in this sentence a verb is missing.

Response 5: We have added a verb “act” as suggested in the revised manuscript.

Point 6: Lines 379-80: "such improvements are made using DNA-free delivery of CRISPR/Cas9...": please add a reference.

Response 6: References have been added as suggested in the revised manuscript.

Additionally, we have double checked the manuscript and changed some words in writing to polish the revised version: “interaction” was changed to “interactions” in Line 280 and “were” to “was” in Line 334, “NBS-LRR” was changed to be italic in writing in  Line 18, Line 140.

Reviewer 2 Report

Authors Yang et al. introduced here a manuscript entitled „miRNA mediated regulation and interaction between plants and pathogens“.

Authors focused their review on miRNA molecules.

They described the role of miRNA molecules in plant disease resistance, through their mediation of expression of disease resistance genes, involving in the phytohormone signaling, involving in reactive oxygen species production.

Further, they mentioned also interactions between disease related miRNA and long non-coding RNAs and regulatory modes of disease-related miRNAs in plants.

Finally, they showed possible applications of miRNAs in molecular breeding for disease resistance.

The review is well written and merits publication.

Author Response

Response to Reviewer 2 Comments

    Authors Yang et al. introduced here a manuscript entitled “miRNA mediated regulation and interaction between plants and pathogens”. Authors focused their review on miRNA molecules. They described the role of miRNA molecules in plant disease resistance, through their mediation of expression of disease resistance genes, involving in the phytohormone signaling, involving in reactive oxygen species production. Further, they mentioned also interactions between disease related miRNA and long non-coding RNAs and regulatory modes of disease-related miRNAs in plants. Finally, they showed possible applications of miRNAs in molecular breeding for disease resistance. The review is well written and merits publication.

    Response: We are very grateful to your comments for our manuscript. We have made some minor changes in the revised version. We have uploaded the revised manuscript using the "Track Changes" in the formats of MS word for your approval.
